# MicroRNA (miRNA): A New Dimension in the Pathogenesis of Antiphospholipid Syndrome (APS)

**DOI:** 10.3390/ijms21062076

**Published:** 2020-03-18

**Authors:** Przemysław J. Kotyla, Md Asiful Islam

**Affiliations:** 1Department of Internal Medicine, Rheumatology and Clinical Immunology, Faculty in Katowice, Medical University of Silesia, 40-635 Katowice, Poland; 2Department of Haematology, School of Medical Sciences, Universiti Sains Malaysia, Kubang Kerian 16150, Kelantan, Malaysia

**Keywords:** MicroRNA, miRNA, antiphospholipid syndrome, APS, autoimmunity

## Abstract

MicroRNAs (miRNAs) are single-stranded, endogenous RNA molecules that play a significant role in the regulation of gene expression as well as cell development, differentiation, and function. Recent data suggest that these small molecules are responsible for the regulation of immune responses. Therefore, they may act as potent modulators of the immune system and play an important role in the development of several autoimmune diseases. Antiphospholipid syndrome (APS) is an autoimmune systemic disease characterized by venous and/or arterial thromboses and/or recurrent fetal losses in the presence of antiphospholipid antibodies (aPLs). Several lines of evidence suggest that like other autoimmune disorders, miRNAs are deeply involved in the pathogenesis of APS, interacting with the function of innate and adaptive immune responses. In this review, we characterize miRNAs in the light of having a functional role in the immune system and autoimmune responses focusing on APS. In addition, we also discuss miRNAs as potential biomarkers and target molecules in treating APS.

## 1. Introduction

MicroRNAs (miRNAs) are a group of small non-coding single-stranded RNA molecules containing approximately 22 nucleotides. The miRNAs are majorly involved in regulating gene expressions [1,2,3]. Since 1993, following the discovery of the first miRNA (*lin-4*), the presence of miRNAs was confirmed in various organisms including humans, animals, plants, and viruses and were found to be evolutionarily conserved across diverse species [4,5]. To date, at least 1917 human miRNA sequences have been identified and registered in the miRBase database of the University of Manchester. miRNAs are predicted to regulate 90% of the protein-encoding genes making them the largest class of gene regulators [6]. miRNAs complimentarily pair with messenger RNAs (mRNAs) and mediate mRNA cleavage, translational repression, and mRNA destabilization. miRNAs are known to be involved in crucial cellular processes, and their dysregulation contributes to the development of a broad range of diseases. Studies have confirmed that miRNAs are the key players in immune cell differentiation and immune responses including antibody secretion and release of inflammatory mediators [7,8]. Therefore, dysregulation of miRNAs contributes to the development of a broad range of diseases associated with autoimmunity, development of cancers, and autoimmune diseases [9,10,11]. Indeed, there are several lines of evidence linking changes in miRNA expression to autoimmune diseases like systemic lupus erythematosus (SLE) [12], rheumatoid arthritis (RA) [13], systemic sclerosis (SSc) [14], multiple sclerosis (MS) [15], Sjögren’s syndrome [12], and antiphospholipid syndrome (APS) [16].

The objective of this review was to summarize the role of miRNAs in appropriate functioning of the immune system with the special emphasis on its disturbances leading to the development of APS. We also discussed the possible miRNA-associated pathophysiological mechanisms that may contribute to the development of a prothrombotic state in APS.

## 2. Search Strategy and Inclusion Criteria

A systematic search was performed to identify studies on miRNAs in APS by using the PubMed, Scopus, and Google Scholar databases. The following combinations of terms were considered: Antiphospholipid Syndrome, APS, Hughes Syndrome, Antiphospholipid Antibody Syndrome, Anti-Phospholipid Antibody Syndrome, Anti Phospholipid Antibody Syndrome, Anti-Phospholipid Syndrome, Anti Phospholipid Syndrome, microRNA, microRNAs, micro RNA, micro RNAs, miRNA, miRNAs, and small temporal RNA.

Articles published in the English language were considered, and the search covered the period until 30th January 2020. The studies focusing on the expression of miRNAs in patients with primary APS and secondary APS compared to normal controls as well as animal models were considered. Studies were included irrespective of the disease progression and treatment methods. Case reports, conference proceedings, editorials, as well as studies without control groups were excluded. In addition, studies where the total number of recruited samples was less than 10 or where complete data were not available were also excluded.

## 3. Origin and Function of miRNAs

There are several places where miRNA-coding genes may originate from. The main places responsible for miRNA coding are localized in the intergenic regions or in antisense orientation of the corresponding genes or intron regions of protein-coding genes [17,18]. Most of the miRNAs are transcribed as independent units; however, when miRNA genes are localized in the intronic regions, the miRNA genes are transcribed as the part of the annotated genes [19].

The first step of miRNA biogenesis occurs in the nucleus to create non-protein-coding RNA transcripts. The key enzyme in this process is RNA polymerase II to create up to one thousand nucleotide long primary miRNA (pri-miRNA) [20]. At least one side of each pri-miRNA is folded back to form a hairpin structure—a substrate for the heterotrimeric enzyme complex commonly referred to as a microprocessor. The essential component of a microprocessor is an endonuclease termed Drosha. Two double-stranded (ds) RNA binding proteins (containing dsRNA-binding domains (dsRBDs)), also known as DiGeorge syndrome critical region gene 8 (DGCR8) microprocessor complex, helps to stabilize the miRNA-endonuclease complex while processing [21]. The DGCR8, sometimes referred to as Pasha (partner of Drosha) in flies and nematodes, is a dsRNA binding protein. The DGCR8 gene in humans is located on chromosome 22q11.2 [22]. Monoallelic deletion of this genomic region is linked with several clinical defects, commonly known as DiGeorge syndrome [23]. It has been shown that DGCR8 depletion resulted in pri-miRNA accumulation and concomitant reduction of precursor miRNA (pre-miRNA) and mature miRNA levels. Even though individually, both the components possess microprocessor activity, neither DGCR8 nor Drosha alone could exert proper microprocessor functioning, indicating that DGCR8 is an essential cofactor for Drosha [21,24]. Drosha has two RNase III domains, and each processes one strand of the stem of the pri-miRNA hairpin liberating a 70–100 nucleotide-long precursor pre-miRNA. The pre-miRNA is then transferred into the cytoplasm by a special carrier, exportin 5/RanGTP, which specifically recognizes the structure of pre-miRNAs [25]. In the cytoplasm, pre-miRNA is cleaved by the complex of Dicer (RNase III family member) and trans-activator RNA binding protein (TRBP) into a 19- to 25-nucleotide miRNA/miRNA* duplex. The miRNA* component is recognized as a passenger strand as it is less stable. The miRNA/miRNA* complex dissociates, and the main strand (guided strand) miRNA is then loaded into RNA-induced silencing complex (RISC) stabilized by argonaute (Ago) and chaperone proteins (HSC70/HSC90) [26]. Once loaded in RISC, the mature miRNA can interact with its target RNA and bind to the 3′-untranslated region (3′-UTR) resulting in degradation or translational repression of the target gene [27]. There are four Ago proteins characterized in mammals (Ago1-4); however, only Ago2 seems to be an active component in the miRNA maturation pathway [28]. The role of Ago2 in this pathway is to cleave mRNA recognized by miRNA, thus working as a catalytic enzyme [29]. Moreover, Ago can modify the properties of its RNA, enabling miRNA binding to the targeted mRNA. When bound to Ago, the so-called “seed region” consists of guide (g) nucleotides g2–g8, pairs with complementary target RNAs 50–200 times faster than the annealing rates of equivalent naked RNAs [30]. When complementary base pairing is perfect or close to perfect, the Ago protein induces deadenylation and degradation of mRNA fragments. In case of incomplete pairing, Ago contributes to dsRNA formation, resulting in translational repression. Inactive dsRNAs aggregate in cytoplasmatic foci referred to as P-bodies [27,31]. The unique function of miRNA is that miRNA can bind to numerous mRNAs, and conversely, various miRNAs can bind to the same mRNA molecule.

## 4. The Role of miRNA in Immune Response

The proper function of the immune system is the key element that protects the organism against dangerous organisms that may potentially invade the host, contributing to its death. The function of the immune system is thus to protect from such dangerous invaders on one side and to prevent recognizing self-structure-antigens and stop or reduce host response against them. The proper function may be therefore characterized as distinguishing self from foreign antigens and mounting the immune response toward non-self-structures. It is not surprising that such a function requires multi-level regulation to avoid recognizing the self-antigens. In cases when such regulation is impaired, autoimmunity develops [32]. In the last twenty years, substantial progress towards understanding the immune mechanism has been made, and it is clear that miRNAs play an important part in this immune regulation [33].

## 5. Innate Immune System

The role of miRNAs in the functioning of the innate immune system is supported by recent findings of miRNA expression in various innate immune system cells such as monocytes, macrophages, dendritic cells (DCs), granulocytes, and natural killer (NK) cells [34,35,36,37]. These cells stay in the first line of defense system and protect the host against various infections. There is strong evidence suggesting that miRNAs play an important role in the development and function of innate immune cells [38].

There are several ways in which miRNAs can regulate TLR-signaling pathways. miRNAs interact with such vitally important processes as TLR expression, TLR regulatory molecule synthesis, expression of TLR-induced cytokines, and TLR-induced transcription factors [39,40].

## 6. Adaptive Immune System

### 6.1. T-Cells

The adaptive immune system mostly comprises B-cells and T-cells, and working together, these cells are able to mount an antigen-specific second line of immune defense. Recent studies have elucidated that miRNAs are critical regulatory factors in the development of key adaptive immune cells—T and B-cells. miRNA can regulate the functions of T-cells in several ways. As the most important miRNA function is to suppress the T-cell response, expression of most miRNA (miR-16, miR-142–3p, miR-150, miR-142–5p, miR-15b, and let-7f) in a naïve T-cell is significantly reduced before T-cell activation [41]. On the contrary, in CD8^+^ T-cells, miR-155 is responsible for T-cell proliferation in response to viral infection [42] and states of miRNA 155 absence, resulting in reduction of antigen-specific CD8^+^ T-cells [43]. The other player in this field is the T helper (Th) cell. Th activity is precisely regulated by miRNA. Similarly to CD8^+^, cell fate of Th is determined by miRNA 155 playing the pivotal role in modulation of immune response. Besides its role in the innate immune compartment and development of CD8^+^ response, miR-155 has been found to promote polarization of T-cells toward Th1/Th17 response during inflammatory conditions. More precisely, the miR-155 inhibiting suppressor of cytokine signaling 1 (SOCS1) in dendritic cells promotes synthesis of proinflammatory cytokines that in turn contribute to Th1/Th17 polarization [44,45]. Interestingly, the same miRNA 155 plays a significant role in the development of regulatory T (Treg) cells. Upregulation of Foxp3 during thymic maturation of Treg resulted in significant increase of miRNA 155 that was preserved in peripheral Tregs. The study of Li-Fan et al. [45], who observed a marked reduction in the proportion and absolute numbers of Foxp3^+^ in miRNA-155-deficient mice, may serve as a confirmation.

The other factor involved in T-cell maturation and function is miRNA 146a. miRNA 146a promotes Th1 differentiation by activation of the STAT1-T-bet axis enabling Th1 polarization [46]. On the other hand, miRNA 146a is deeply involved in Treg functioning and reduction of the development of Th17 cells [47].

### 6.2. B-Cells

A large and growing body of evidence indicates that B-cell function is regulated by miRNAs. To date, several miRNAs have been reported that may act at several stages of B-cell development, maturation, and function [48,49]. Six mature miRNAs (miR-17, miR-18a, miR-19a, miR-20a, miR-19b-1, miR-92a-1, and two paralogs miR-106a-362 and miR-106b-25) referred to commonly as the miR-17-92 cluster are important for B-cell development, being responsible for such processes as B-cell differentiation at pro-B to pre-B stage [50]. Moreover, animals lacking a miRNA cluster die shortly after birth due to serious internal organ malformations [48]. Another factor involved in B-cell lymphopoiesis is miR-150, which reduces maturation of B-cells at the level of pro- to pre-B transformation [51].

The mir-23a cluster consists of three miRNAs, namely miR-23a, miR-24, and miR-27a. The main function of this cluster is to suppress B-cell maturation and provide an equilibrium between B and myeloid cells [52]. However, the role of miR-181a goes far beyond B-cell regulation, and it is deeply involved in such processes as gastric cancer, and lymphoblastic leukemia or may serve as a biomarker in multiple myeloma [53,54,55].

## 7. MicroRNA in Autoimmunity

Several lines of evidence exist regarding the role of miRNA in the development of autoimmune diseases. Recently in 2020, Zhang et al. [56] in their meta-analysis identified several miRNAs that play significant roles in the progression of more than one autoimmune disease. Among them, miR-21, miR-26a, miR-155, miR-148a, and miR-223 seem to play crucial roles. These miRNAs are deeply involved in processes like immune cell-apoptosis suppression and specific organ injury including the skin, joint, lung, liver, and kidneys. It is not surprising that while these miRNAs are commonly associated with immune response, any changes of their expression may lead to increased activity of the immune system [57,58]. Specific profiles of hundreds of autoimmune disorder-associated miRNAs have been identified so far. Many of the miRNAs may serve as diagnostic or prognostic biomarkers. For instance, in SSc and RA, MiR-30a suppresses the synthesis of B-cell activating factor (BAFF), thereby affecting the survival of B-cells [59]. Considering the role of B-cells in progression of SSc, it may be speculated that suppression of miR-30a may influence the development of SSc [60]. Moreover, recent data suggest that specific miRNA signature enables distinction between various connective tissue diseases [61].

Even more data have been accumulated in regard to SLE. Many recent findings illuminate transcriptional regulation and chromatin modification as the critical point for development of autoimmunity, including connective tissue diseases [62,63,64,65,66,67]. Following the first study in 2007, hundreds of miRNAs have been identified as having a potential role in modulating the immune system toward autoimmune response. Interestingly, apart from many discrepancies existing in this field, some recent data exhibited that a limited number of miRNAs (i.e., miR-181, miR-186, and miR-590-3p) may target at least 50% of the lupus-related genes and be associated with disease predisposition [68].

## 8. Antiphospholipid Syndrome

APS is a systemic autoimmune systemic disease, affecting multiple organs and characterized by arterial and venous thromboses and/or obstetric morbidity (i.e., recurrent fetal loss) [69,70]. The formal classification includes at least one clinical criteria in the presence of the persistent positivity of at least one of the antiphospholipid antibodies (aPLs) [71,72]. Laboratory criteria include presence of aPLs (i.e., lupus anticoagulants (LA), anticardiolipin (aCL) IgG or IgM, and/or anti-β2-glycoprotein I (β2GPI) IgG or IgM) on two or more occasions at least 12 weeks apart and no more than 5 years prior to clinical manifestations. Currently, several variants of the disease are recognized including primary, secondary (concomitant with other autoimmune diseases), and the most severe, referred to as catastrophic APS. APS patients are characterized by a wide spectrum of clinical manifestations such as livedo reticularis, migraine, epilepsy, thrombocytopenia, valvular heart disease, hemolytic anemia, and nephropathy. These clinical presentations, although frequent, are not included in classification criteria [73,74,75].

The serological hallmark of the disease is the presence of aPLs, antibodies that target phospholipids or phospholipid-binding protein complexes. Among many antigens proposed so far, β2GPI seems to play the crucial role as it has been identified as the main autoantigen in APS. Apart from β2GPI, in the special settings of APS the other antigens may become the target for aPLs [76]. Among them, phosphatidylserine, thrombomodulin, annexin A5 and A2, and kininogens like protein C and S are most often studied in the light of APS pathophysiology. aPLs bind to negatively charged β2GPI expressed on cellular surfaces of endothelial cells, platelets, and monocytes polarizing toward prothrombotic phenotype and subsequent complement cascade activation [77]. The other mechanisms playing a role in this field are tissue factor (TF) expressed by neutrophils, neutrophil extracellular traps (NETs), and the upregulation of the mechanistic target of rapamycin (mTOR) complex on endothelial cells. This procoagulation storm may bring many pathophysiological consequences as the deep interaction is stable coagulation—the anticoagulation state results in several hemostatic reactions, such as acquired protein C resistance, inhibition of TF pathway inhibitor, or inhibition of tissue plasminogen inhibitor activity [78].

The prevalence of aPLs is estimated to be approximately 1%–5% in the general population; however, it has the tendency to increase with older age. APS is a disease that affects women more often than men, and the male-to-female ratio ranges between 1:3.5 for primary APS to 1:7 when disease is developed in the course of the other connective tissue diseases (specifically SLE, but also SSc, Sjögren’s syndrome, dermatomyositis, and RA) [69,79]. Although aPLs’ positivity is the hallmark of APS, aPLs can be present in various pathological states such as infections and malignancies [80,81]. In such cases, the titers of aPLs are generally low and transient, and the role of them in thrombotic event development is uncertain [69]. Contrary to this, aPLs’ persistence was shown to increase the annual risk of thrombosis by up to 5%. The highest risk is observed in patients with so-called triple positivity, which refers to the state when all the three aPLs are present simultaneously (LA, aCL, and anti-β2GPI) [82].

## 9. APS: Genetic Predisposition and Family Studies

Although the precise pathogenic background of the syndrome is unknown, in some cases genetic predisposition may play a role. In the recent systematic review, 16 potential genes participating in the development of APS were identified [83]. It was 40 years ago when the first familial occurrence of aPLs was documented for the first time [84]. Following this finding, several studies identified families affected by APS [85,86,87].

The most obvious link was human leukocyte antigen (HLA) genes, and this relationship has been tested since the beginning. Indeed, some published papers underlined associations between APS and HLA. With the help of molecular methods, it has been established that HLA-DQw7 (DQB1*0301 allele) is linked to APS [88]. This work was further substantiated by the study of Asherson et al. [89] who discovered increased expressions of DR4 and DRw53 in patients parallel to the absence of DR3 in all patients where both class II genes and class III genes were examined by molecular methods. They found that significant differences were limited to the HLA class II region of the MHC. At the beginning of this century, in a cohort of 53 Caucasian APS patients in the UK, Caliz et al. [90] established that the haplotypes DQB1*0301/4-DQA1*0301/2-DRB1*04 and DQB1*0604/5/6/7/9-DQA1*0102-DRB1*1302 were more frequent in patients with primary APS than in controls. The similar conclusions on the role of the MHC come from studies from Mexico [91], the UK [92], and the United Arab Emirates [93]. All studies showed a close link between APS and expression of several HLA genes. The same conclusion came from studies considering secondary APS (mainly related to SLE) as well. In several published studies the role of such HLA antigens such as C4Q0, DR4, DR7, DRw53, DRB1*1401,0301, DRB1*0402/3, DRB1*07, DQA1*0201, DQA1*0301 and DQB1*0302 has been underlined [94]. The genetic predisposition is obviously not restricted to HLA genes alone. Several other factors involved in blood clotting should be taken into consideration; among them factor V Leiden and G20210A prothrombin polymorphism have attracted attention [95].

## 10. MicroRNA and Antiphospholipid Syndrome

miRNAs have a strong impact on immune response; therefore, it is not surprising that those small particles regulate the formation of aPLs or that, on the contrary, aPLs may modulate miRNAs. Despite the mechanisms involved, in both situations, disease-specific antibodies possess the ability to influence the final shape of the disease. However, contrary to SLE and other connective tissue diseases, there are limited data on direct role of miRNA in APS (Table 1).

The role of miRNA in APS was recognized for the first time in the second decade of this century (Figure 1). Teruel et al. [96] in their pioneering study identified the factor that may modulate TF expression. Specifically, they demonstrated that miR-19b and miR-20a, two particles belonging to the miR-17-92 cluster, may modulate TF expression in monocytes of patients with APS and SLE. On the other hand, several studies have shown that aPLs induce the TF formation in monocytes and endothelial cells, thus promoting hypercoagulable state [99,100,101]. In a study [96], it was shown that expressions of miR-19b and miR-20a were significantly reduced in patients with APS and SLE (approximately 30% as compared to healthy controls). The authors suggested that low levels of miR-19b and miR-20a may translate at least indirectly to higher expression of TF. This is not a surprising finding as miR-19b and miR20a directly bind to TF mRNA and suppress mRNA translation. This model gives broader insight into mechanisms of the procoagulation state seen in APS. The probable scenario is that TF regulating miRNAs are reduced in patients as the result of genetic background, parallel to increased expression of TF induced by aPLs. The still open question is what the potential mechanism is that is responsible for reducing miRNA expression. Several hypotheses have been coined so far including single nucleotide polymorphism (SNP) in SLE and APS patients, gain or loss of miRNA function, or differentially expressed miRNA in the disease [102]. TF expression is obviously one among many mechanisms responsible for procoagulation state in lupus. Recently, it has been shown that another miRNA, mir-133a, influences vitamin K 2,3-epoxide reductase complex subunit 1 (VKORC1) [103]. This is an essential element involved in the correct γ-carboxylation of vitamin-K-dependent proteins such as Gas6, matrix-GLA, and osteocalcin, as well as hemostatic proteins C, S, and Z and coagulation factors II, VII, IX, and X [104,105].

The role of miR-133a goes far beyond the influence on VKORC1 alone as this molecule has been shown to play a role in such pathological states as atherothrombosis and myocardial infarction [106,107]. The direct role in APS, however, remains to be elucidated. It may be only speculated that since mi133a is IL-19 dependent, and plays a protective role in atherothrombosis, any shift toward pro-inflammatory response and reduction of IL-19 (and the other IL-10 family members) may contribute to procoagulation state [107]. The same effect of miRNA regulation of IL-10 dependent immune response has been observed by a group from Israel. In an APS mouse model, elevated circulating miRNA 98 has been observed, but this effect was reversed by oral tolerogenic diet of D1-β2GPI [108]. These findings provide the direct link between miRNA 98 and APS and directly show that miRNA 98 expression is the result of immune system activity. Of note is that miRNA 98 has been shown to negatively regulate IL-10 anti-inflammatory cytokine expression by macrophages [109]; thus, any reduction of miRNA 98 may increase immune response and APS activity. In a similar study, the same group showed that immunization with β_2_GPI-domain-I derivative contributes to the formation of specific tolerogenic dendritic cells (β2GPI D-I tDCs), which in turn possess the ability to attenuate APS in a murine model. Moreover, the tolerance induced by β2GPI tDCs, correlated with the enhanced expression of T-reg cells, amelioration of proinflammatory cytokines IL-1β and IL-17 levels, and upregulation of anti-inflammatory IL-10 and transforming growth factor (TGF)-β expression [110]. The initial finding of Teruel et al. was further substantiated by the findings of Perez-Sanchez et al. [98], who identified miRNAs potentially targeting mRNA involved in the development of atherothrombosis, immune responses, oxidative stress, and intracellular signaling related to APS. The most important miRNAs characterized in the study were miR-124a-3p, miR-125a-5p, miR-125b-5p, miR-146a-5p, miR-155-5p, and miR-222-3p. Moreover, the lower expression of all these miRNAs were found in neutrophils of patients with APS and SLE in comparison to heathy controls. Additionally, miR-124a and miR-125a were reduced in monocytes of APS and SLE patients, while miR-146a and miR-155 were significantly increased.

Another study conducted by the same group identified 39 miRNAs differentially expressed in patients with APS compared to healthy subjects. Among them the authors identified 19 upregulated and 20 downregulated miRNAs involved in connective tissue diseases, immune response, vascular diseases, and thrombosis [16]. Moreover, APS patients showed a specific miRNA profile different from aPL-negative subjects with SLE and non-immune patients with previous thrombotic events. The miRNA expression did not change over the time, apart from the miR-20a/374 ratio, and standard treatment has no effect on miRNA expression. The study helped to identify specific miRNA expressions and confirmed APS as a distinct clinical entity in spite of many clinical similarities to SLE and thrombotic disorders. In the second part of the study, the authors used Ingenuity Pathway Analysis (IPA) to identify potential mRNA targets involved in clinical features of APS. This method allowed the identification of 11 altered miRNAs as the main regulators of proteins involved in the pathological background of APS, among them miRNA 34a-5p, 15a-5p, 145a-5p, 133b-3p, 124-3p, 206, 20a-5p, 19b-3p, 210-3p, 296-5p, and 374a-5p. In this set of 11 miRNAs, the top 5 upregulated miRNAs and 3 out of top 5 downregulated miRNAs were characterized in the PCR-array. Finally, the expression levels of those miRNAs were analyzed, revealing that miR-124 and miR-34a were increased in APS patients as compared to healthy subjects, while miR-20a, miR-19b, and miR-145a were reduced [16].

Of note is the fact that aPLs can modulate the expression of circulating miRNAs, which in turn (at least indirectly) promote the secretion of atherothrombotic proteins such as TF, plasminogen activator inhibitor (PAI-1), and MCP-1 [111]. miRNA 124, which modulates activity of monocyte chemoattractant protein (MCP)-1, miRNA 133b, and miRNA 145 playing the role in differentiation of vascular smooth cells and involved in the pathogenesis of atherosclerosis may serve as examples [112,113,114,115]. Using the miRNA target filter, the authors created a self-related miRNA-target network, linking the expression of APS-related microRNAs to such disorders as cardiovascular, thrombosis, and cerebrovascular (presented in reference [16]).

The role of miRNA, however, should be recognized in a wider context, not only as molecules that possess the ability to change expression of several signaling molecules and receptors but most of all as the molecules that directly orchestrate APS-specific immune response. In other words, they regulate the key steps in the development of the diseases and contribute to its final shape (i.e., clinical presentation, severity, and response to the treatment). Three important factors should be taken into consideration. Firstly, the permissive background, which is discussed in detail in the first paragraph, should be considered. The second factor, postulated for decades, is the so-called “second hit”. The nature of second hit is still obscure, but the general agreement is that it might be a result of inflammation, cellular stress, infections, and many others [76]. The point is that in spite of the nature of the second hit, it provides signals that may be recognized by pattern-recognition receptors such as TLRs. When stimulated, it recruits several adaptor proteins and activates many pathways resulting in activation of several genes involved in immune response (both in activation but also suppression of some of them) [116]. The most important immune pathways consist of JAK-STAT, PI3K-Akt-mTOR, and MAPK/NFκB pathways [116]. Thirdly, all steps of vitally important inflammatory and immune pathways can be regulated at the level of miRNA [39,117]. Quite recently, it has been postulated that aPLs (specifically anti-β2GPI antibody) may act as a PAMP, and they were observed to stimulate directly inflammatory pathways through TLR4 [118,119,120]. Again, this process can be regulated by miRNAs, precisely, at almost all levels of signaling pathways [121,122]. Moreover, the other inflammatory pathways, namely CD40 and IL-1 receptor, share some common adaptor proteins like TNF receptor-associated factor 6 (TRAF6) [123,124]. The next step in this common pathway is to activate NFκB and activator protein (AP)-1, which leads to TF expression, thus promoting hypercoagulability as is observed in APS [120,125]. Interestingly, some aspects of APS, namely endothelial cell proliferation, invasion, and migration, can be suppressed by 27nt-miRNA targeting AP-1, which may serve as an example of multi-level regulation of transcriptional factors [126].

Less is known on the role of miRNA in regard to clotting factors in APS. In spite of the well-known role of miRNAs’ regulation of TF, the role of microRNAs in most of clinical presentation of APS can be only reviewed indirectly from studies in other pathological states as venous deep thrombosis [127,128], severe trauma-related hypercoagulation state [129], prethrombotic states in coronary vascular disease [130], thrombosis- inflammation response [131,132], or hemophilia.

## 11. Future Direction

In spite of growing evidence on the role of miRNA in development and progression of APS, direct data in this field are scant. Most studies addressed secondary APS, and this approach is not free from bias, as it is not possible to precisely distinguish whether changes of miRNAs are due to APS alone or due to the influence of SLE. Even huge APS registries address primary and secondary APS together. The prevalence of APS is relatively low—another fact that potentially challenges the final conclusions. Although the number of published papers on APS and miRNA is really low, several upregulated and downregulated miRNAs have already been identified so far that have influenced a few important conclusions. Further assessment of these miRNAs may create unique mosaics including risk stratification in different clinical settings (i.e., thrombotic events, pregnancy morbidity, triple aPLs positive) of APS and potential biomarkers in disease progression. To go further, modulation of miRNA expression may cease even reverse procoagulatory status of APS patients, and such an attempt has already been undertaken in tolerogenic studies from Israel [110], showing it to be a promising way to modulate inflammatory response in APS. There are thousands of miRNA molecules waiting to be discovered, which may potentially link with the pathogenesis of APS. However, prior to that, we need to understand clearly the roles of the confirmed miRNAs in APS. This may open new ways to modulate the activity of the disease and reveal the risks of developing APS, which may directly translate to everyday clinical practice. We do believe that understanding the role of confirmed miRNAs so far may give the strong stimulus for further studies, which we do hope will contribute extensively to describing the roles of miRNAs in APS.

## 12. Conclusions

There are several factors that may potentially link miRNAs and APS. Firstly, APS is present in the course of connective tissue diseases, where the presence of aPLs is not only a marker of the disease but also is directly responsible for all pathophysiological events observed in the patients. Secondly, multi-level regulation of all crucial steps in immune response and signaling pathways by miRNAs may be an important factor. Finally, lower expressions of most regulatory miRNAs have been found that may be recognized as a potential permissive background. Understanding as to why dysregulation of miRNA expression is present may open the way to create new therapeutic approaches. What is already known is that treatment with immunosuppressants changes the expression of several potent miRNAs; thus, the treatment of APS immunosuppressants should be considered as an anchor therapy together with anticoagulants and antiplatelet drugs.

## Figures and Tables

**Figure 1 ijms-21-02076-f001:**
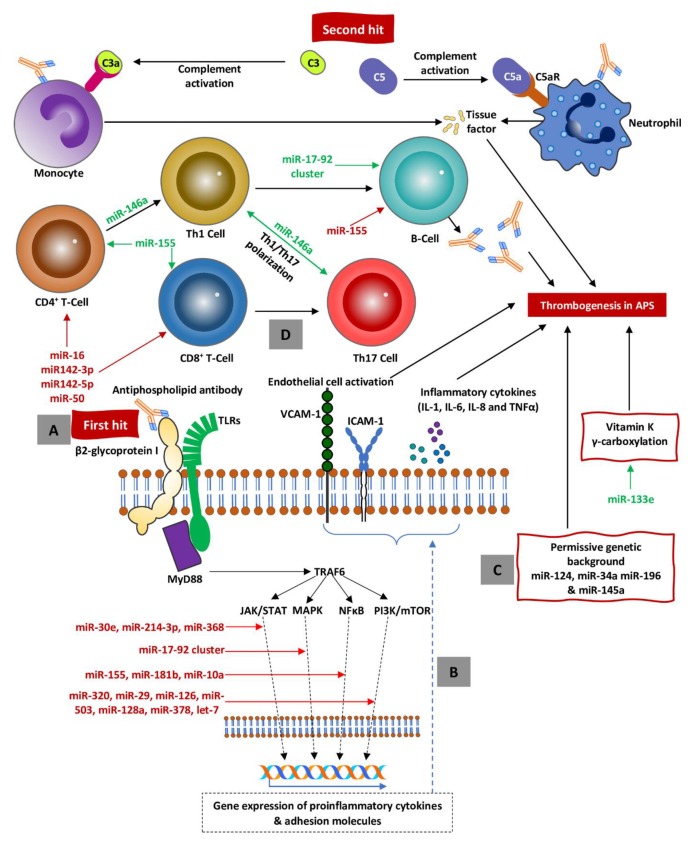
**A.** A partially unknown stimulus referred to as a first hit (i.e., DAMPs, PAMPs, and aPLs) stimulates TLRs, which in turn activate signal pathways resulting in activation of proinflammatory cytokine genes. **B.** As the result of inflammatory gene activation, synthesis of inflammatory cytokines and expression of adhesion molecules (i.e., ICAM-1 and VCAM-1) on the endothelium surface occurs. **C.** Permissive background creates procoagulatory milieu to perpetuate a shift towards coagulation and endothelial damage. **D.** As the result of suppression of regulatory miRNA (marked in red), the immune system is shifted towards active proinflammatory response with Th polarization toward Th1 and Th17 responses, subsequent activation of B-cells, and autoantibody secretion (including aPLs) causing inflammatory storm (activation of endothelium and clothing factors). All the crucial steps are precisely regulated at the level of miRNAs, which mainly suppress (red text) but also (to a lesser degree) activate (green text) procoagulation. Abbreviations: miRNAs = microRNAs, APS = antiphospholipid syndrome, PAMPs = pathogen-associated molecular patterns, DAMPs = danger-associated molecular patterns, TLRs = toll-like receptors, TRAF6 = tumor necrosis factor receptor-associated factor 6, JAK/STAT = janus kinases/signal transducer and activator of transcription, NF-κB = nuclear factor kappa-B, MAPK = mitogen-activated protein kinase, mTOR = mechanistic target of rapamycin, aPLs = antiphospholipid antibodies.

**Table 1 ijms-21-02076-t001:** Changes in miRNA expression between primary and secondary antiphospholipid syndrome (APS) and healthy controls.

miRNA	Type of Cells	Target (or Population Studied)	Activity	References
miR-19b	White blood cells	Tissue factor	Downregulated	[96]
miR-20a				
miR-296-5p	Plasma and supernatants	Tissue factor, Plasminogen activator inhibitor, Monocyte chemoattractant protein, Vascular endothelial growth factor	Upregulated	[16]
miR-133b				
miR-124-3p				
miR-206				
miR-34a-5p				
miR-423-5p				
miR-122-5p				
miR-193a-5p				
miR-210-3p				
miR-192-5p				
miR-25-3p				
miR-204-5p				
miR-31-5p				
miR-205-5p				
miR-150-5p				
miR-196a-5p				
miR-885-5p				
miR-155-5p				
miR-373				
miR-20a-5p			Downregulated	
miR-30d-5p				
miR-24-3p				
miR-17-5p				
miR-30a-5p				
miR-19b-3p				
miR-191-5p				
miR-128-p				
miR-106b-5p				
miR-22-3p				
miR-26a-5p				
miR-26b-5p				
miR-376c-3p				
miR-222-3p				
miR-103a-3p				
miR-15a-5p				
miR-211-5p				
miR-145-5p				
miR-374a-5p				
miR-143-3p				
miR-125b	Dendritic cells	PAPS vs. HC	Downregulated	[97]
miR-127a		PAPS vs. HC		
miR-150a		SLE+APS vs. HC		
miR-181 a		PAPS vs. HC		
miR-221a		PAPS vs. HC		
miR-335		PAPS vs. HC		
miR-362		SLE+APS vs. HC		
miR-532		SLE+APS vs. HC		
miR-29a		SLE+APS vs. HC PAPS vs. HC		
miR-196b		SLE+APS vs. HC		
let-7g		PAPs vs. HC		
miR-744		PAPs vs. HC		
miR-193b		SLE+APS vs. HC		
let-7e		PAPs vs. HC		
miR-30a-5p		PAPs vs. HC		
miR-30d		SLE+APS vs. HC		
miR-30e-3p		SLE+APS vs. HC PAPs vs. HC		
mir590-3p		PAPS vs. HC		
miR-126		PAPS vs. HC		
miR-1275		SLE+APS vs. HC		
miR-4443	Neutrophils	SLE and APS vs. HC	Upregulated	[98]
miR-146b-5p				
miR-302d-3p				
miR-7-5p				
miR-193a-5p				
miR-320e			Downregulated	
miR-346				
miR-155-5p				
miR-22-3p				
miR-486-3p				
miR-15a-5p				
miR-144-3p				
miR-186-5p				
Let-7g-5p				
miR-151a-3p				
miR-32-5p				
miR-27b-3p				
miR-548aa				
miR-194-5p				
miR-4431				
miR-21-5p				
miR-324-5p				
miR-374a-5p				
miR-132-3p				
miR-126-3p				
miR-450-5p				
miR-140-5p				
miR-494				
miR-301a-3p				
miR-142-3p				
miR-92a-3p				
miR-30e-5p				
miR-590-5p				
miR-339-3p				
miR-630				
miR-71-5p				
miR-106b-5p				
miR-1537				
miR-197-3p				
miR-503				
miR-582-3p				
miR-340-5p				
miR-27a-3p				
miR-26b-5p				
miR-338-3p				
miR-30b-5p				
miR-1260b				
miR-302b-3p				
miR-484				
miR-532-5p				
miR-4454				
miR-26a-5p				
miR-17-5p				
miR-299-3p				
miR-29b-3p				
miR-125-5p				
miR-875-5p				
miR-142-5p				
miR-363-3p				

Abbreviations: APS = Antiphospholipid syndrome, PAPS = Primary APS, SLE = Systemic lupus erythematosus, HC = Healthy control.

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
