# Peer review of "MicroRNA (miRNA): A New Dimension in the Pathogenesis of Antiphospholipid Syndrome (APS)"

_ijms, 2020, doi:10.3390/ijms21062076_

Round 1

Reviewer 1 Report

I have read with interest the manuscript titled "MicroRNA (miRNA): A new dimension in the pathogenesis of antiphospholipid syndrome (APS)" by Przemyslaw Kotyla and Asiful Islam.

The manuscript is worthy of interest and try to address a very complex and emerging area of basic and translational research in the field of autoimmunity.

Although the attempt of the Authors is commendable, some major points need to be addressed:

1.The paragraph structure of the paper is clear and neat, and the methods well explained, but the overall balance needs to be adjusted according to the main focus of the manuscript which should be APS. Therefore, the introduction section on the role of miRNAs in immune system and autoimmunity, although interesting, needs to be limited and shortened.

2.While the first part of the manuscript appears to be well written, major editing of English language (preferentially done by a native speaker) is needed in the second part of the paper.

3.APS is not considered a connective tissue disease. It is an autoimmune systemic disease characterized by the persistent positivity for aPL, recurrent pregnancy morbidity and/or thrombotic events, both venous and arterial. Please review the APS paragraph from a clinical perspective since several incorrect information has been noted.

4. Data on the role of miRNAs in APS setting are very limited. Nevertheless, it would be of interest to address some points which could add original value to the manuscript: why data on this issue are still lacking (critical evaluation, for instance: rare and low prevalence disease? methods of research?); possible translational application of miRNAs in clinical practice (from a prognostic, diagnostic, risk stratification assessment, outcome, treatment monitoring and prognosis); possible use of miRNA for therapeutic purposes?; future steps and perspectives?

Author Response

Comments of Reviewer 1

  1. The paragraph structure of the paper is clear and neat, and the methods well explained, but the overall balance needs to be adjusted according to the main focus of the manuscript which should be APS. Therefore, the introduction section on the role of miRNAs in immune system and autoimmunity, although interesting, needs to be limited and shortened.

Response of the authors

Many thanks for your appreciation and insightful comments. We agree with the reviewer that it will read better if we reduce the length of basic information (i.e., role of miRNA in the immune system and autoimmunity) keeping the focus on APS and miRNA. Due to the fact, we have shortened the length of the revised manuscript keeping our focus more on APS and miRNA. In total, we have reduced 689 words (2.5 double space pages). Hope it’s alright now.

  1. While the first part of the manuscript appears to be well written, major editing of English language (preferentially done by a native speaker) is needed in the second part of the paper.

Response of the authors

Thank you for the suggestion. We have considered your comment very seriously and sent the revised manuscript to MDPI English editing service. We hope that in terms of English, the revised manuscript is alright now. See attached certificate for your reference

  1. APS is not considered a connective tissue disease. It is an autoimmune systemic disease characterized by the persistent positivity for aPL, recurrent pregnancy morbidity and/or thrombotic events, both venous and arterial. Please review the APS paragraph from a clinical perspective since several incorrect information has been noted.

Response of the authors

We apologise for the unintentional mistake. We have carefully reviewed the paragraph and corrected already. Hope this paragraph reads better now. Thank you.

  1. Data on the role of miRNAs in APS setting are very limited. Nevertheless, it would be of interest to address some points which could add original value to the manuscript: why data on this issue are still lacking (critical evaluation, for instance: rare and low prevalence disease? methods of research?); possible translational application of miRNAs in clinical practice (from a prognostic, diagnostic, risk stratification assessment, outcome, treatment monitoring and prognosis); possible use of miRNA for therapeutic purposes?; future steps and perspectives?

Response of the authors

We do appreciate your suggestion on discussing the issue critically and therefore, we have introduced a new subheading “Future direction” under which we have discussed critically as per the suggestions of the reviewer. Hope this part adds value to the revised manuscript. Thank you.

Reviewer 2 Report

This review article is comprehensive and informative.

Author Response

Comments of Reviewer 2

This review article is comprehensive and informative.

Response of the authors

We do highly appreciate your appreciation and overall excellent evaluation score. Many thanks.

Round 2

Reviewer 1 Report

Thank you for your careful review. I think that the manuscript has improved significantly. 

Minor comments: 

  1. “The formal classification requires presence of at least one clinical symptom and the presence of any of the three circulating antiphospholipid antibodies (aPL)”. I would suggest rephrasing this sentence as follows: “The formal classification includes at least one clinical criteria in the presence of the persistent positivity of at least one antiphospholipid antibodies (aPLs) (REF)
  2. “Patients with APS are characterized by a wide spectrum of symptoms..”. I would change this sentence as follows “APS patients are characterized by a wide spectrum of clinical manifestations, such as..”
  3. “..and such an attempt has already been undertaken in tolerogenic studies from Israel”. Please add a reference.  

Author Response

We appreciate all suggestions, and followed them in our manuscript

  1. lines 194 and 195 , description of APS has been rephrased as proposed
  2. lines 200 we followed advice and rewrote the APS characteristics as suggested
  3. line 399 the reference number was added to the manuscript 

We would like to express our thanks for all suggestions and comments which contributes enormously to the better shape of our manuscript